# EAGLE: ENHANCED VISUAL GROUNDING MINIMIZES HALLUCINATIONS IN INSTRUCTIONAL MULTIMODAL MODELS

## ABSTRACT

Large language models and vision transformers have shown impressive zero-shot capabilities, enabling significant transferability in downstream tasks. The fusion of these models has resulted in multi-modal architectures with enhanced instructional capabilities. Despite incorporating vast image and language pre-training, these multi-modal architectures often generate responses that deviate from the ground truth in the image data. These failure cases (false positives) are known as hallucinations. Current methods for mitigating hallucinations generally focus on regularizing the language component, improving the fusion module, or ensembling multiple visual encoders to improve visual representation. In this paper, we address the hallucination issue by directly enhancing the capabilities of the visual component. Our approach, named EAGLE, is fully agnostic to the LLM or fusion module and works as a post-pretraining stage that improves the grounding and language alignment of the visual encoder. We show that a straightforward reformulation of the original contrastive pre-training task results in an improved visual encoder that can be incorporated into the instructional multi-modal architecture without any additional instructional training. Extensive empirical validation shows that EAGLE significantly reduces hallucinations across six different instructional multi-modal models and four challenging benchmarks.

## 1 INTRODUCTION

Large-scale pre-trained models have significantly advanced the fields of Natural Language Processing (NLP) and Computer Vision. In NLP, Large Language Models (LLMs) demonstrate strong zero-shot performance on multiple tasks Brown et al. (2020); Devlin et al. (2018); Liu et al. (2019); Raffel et al. (2020); Chiang et al. (2023); Chung et al. (2024); Touvron et al. (2023), shifting research from task-specific to task-agnostic paradigms. In parallel, Vision Transformers (ViTs) Dosovitskiy et al. (2020) have achieved similar breakthroughs in visual understanding, enabling effective zero-shot transfer to downstream tasks Radford et al. (2021); Sun et al. (2023); Oquab et al. (2023); Zhai et al. (2023); Chen et al. (2020); Grill et al. (2020); He et al. (2020); Cherti et al. (2023).

Building upon the zero-shot capabilities of the vision and language models, Instruction Tuning Vision and Language Models (IT-VLM) have emerged as the ensemble of an LLM and a large-scale ViT through a small fusion sub-network known as adapter or connector Li et al. (2023a); Liu et al. (2023a); Dai et al. (2024); Zhu et al. (2023); Peng et al. (2024); Bai et al. (2023); Dong et al. (2024); Research (2024). IT-VLMs achieve impressive zero-shot performance on complex multimodal tasks such as visual question answering, captioning, and grounding. Despite these successes, IT-VLMs often produce erroneous outputs unrelated to image ground-truth, these false positive errors are known as *hallucinations* Villa et al. (2024); Li et al. (2023b); Tong et al. (2024); Guan et al. (2024). This phenomenon is often the result of language biases in the decoder module, as the decoding component would generate outputs consistent with the current stream of generated tokens while ignoring the semantic information in the visual modality Villa et al. (2024).

Existing methods for reducing hallucinations in IT-VLMs primarily focus on the language component (LLM) or the adapter module. These methods include optimizing the adapter training strategy Peng et al. (2024); Bai et al. (2023), improving the adapter module's architecture Li et al. (2023a);

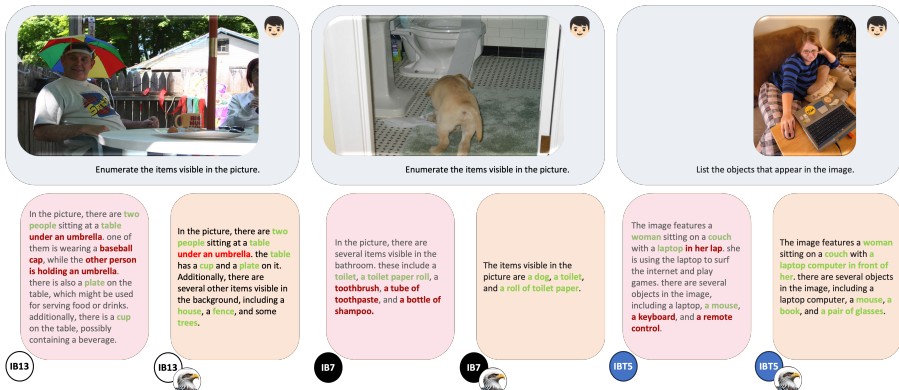

Figure 1: **EAGLE visual encoders reduce hallucinations in IT-VLMs.** We present three example scenarios, each featuring a language prompt and an image. This tuple is input to an IT-VLM with its original visual encoder (left pink box) and the corresponding EAGLE-tuned visual encoder (right orange box). "IB7", "IB13" and "IBT5" refer to InstructBLIP with Vicuna7B, Vicuna13B, and FlanT5xl, respectively. EAGLE substantially reduces hallucinations across different LLMs, providing more visually grounded and reliable descriptions. **i)** In the first example (left), IB13 with the EAGLE-tuned visual encoder can correctly identify fine-grained elements such as a fence, house, and trees. **ii)** In the second example (center), EAGLE helps IB7 to accurately recognize a dog, even in an unusual viewpoint and context. **iii)** In the third example (right), EAGLE enhances object localization, allowing IBT5 to precisely identify the laptop's position. These examples illustrate EAGLE's effectiveness in improving visual grounding across complex, multi-object scenes.

Dai et al. (2024); Liu et al. (2023a), leveraging extensive data to pre-train robust LLMs Chen & Wang (2022); Bai et al. (2023); Liu et al. (2023a), and enhancing instructional training data through commercial LLMs Liu et al. (2023a;b). While effective to some extent, these approaches often overlook the critical role of the visual representation in enabling grounded language generation.

In this work, we take an orthogonal approach: we address hallucinations by directly enhancing the visual encoder. Prior approaches to improve the visual representation Tong et al. (2024); Kar et al. (2024) introduce additional visual encoders, which increase model complexity, hinder scalability and require further fine-tuning of the IT-VLM. We depart from that practice and develop a method that directly improves the ViT ability to encode fine-grained objects details into the image feature sequence while preserving its global image descriptor (CLS). Crucially, our augmented ViT seamlessly integrates into an IT-VLM model without any additional fine-tuning or adaptation. In Figure 1, we provide some qualitative examples that show the baseline (left pink box) and improved response (right orange box) of IT-VLMs. Notably, when our ViT is incorporated, we observe significantly more factual and visually grounded responses.

In this paper, we introduce **E**nhanced Visu**A**l **G**rounding Minimizes Hallucinations in Instructional Mu**L**timodal Mod**E**ls (EAGLE), a post-pretraining strategy that significantly reduces the hallucinations in IT-VLMs. Unlike other methods, EAGLE does not target a specific IT-VLM architecture, or a particular LLM. In fact, EAGLE is fully agnostic to the other IT-VLM components. At training time, EAGLE optimizes the ViT component in isolation from the LLM and fusion modules. At inference time, the improved visual transformer replaces the original ViT of the IT-VLM, no further modification or training is required. Without any bells and whistles, our tuned ViTs reduce the hallucinations across six different IT-VLMs and four well-established benchmarks.

**Contributions.** Our contributions are two-fold: **i) Enhancing Fine-Grained Visual Representation in VLMs.** We show that EAGLE substantially improves the visual representation quality of ViTs within VLMs, enabling them to capture fine-grained visual details in the image feature sequence. Across three state-of-the-art VLMs, EAGLE delivers up to 13.33% higher accuracy on the challenging MMVP-VLM benchmark while preserving zero-shot performance. **ii) Hallucination Reduction without Instructional Training.** By simply replacing the original visual encoder with a EAGLE-enhanced encoder, we mitigate hallucination in IT-VLMs without any additional instruction training. Our method achieves gains of 8.67% on MMVP and 3.23% on MERLIM, and reduces hallucination on AMBER by up to 6.70% across six IT-VLMs and five different LLMs, demonstrating strong generalization. Notably, EAGLE surpasses the visual-ensemble approach of Tong et al. (2024) by 6.00% on MMVP, underscoring its effectiveness in capturing detailed visual information.

## 2 RELATED WORK

Hallucinations in IT-VLMs, are erroneous or ungrounded elements generated during multimodal inference. These events pose significant challenges to the reliability and trustworthiness of these muti-modal systems. Hallucinations often arise from the model reliance on spurious correlations between visual and textual modalities, inherent language biases within the language model component Zhou et al. (2024), a tendency to prioritize linguistic cues over visual information Guan et al. (2024); Li et al. (2023b), and limitations within the visual encoder's capacity to capture fine-grained visual details Tong et al. (2024); Villa et al. (2024).

Existing strategies for mitigating IT-VLM hallucinations focus mainly on curating and collecting more instructional data, improving multimodal alignment, and improving visual grounding.

**Instructional Data Deficiencies.**   Prior works Dai et al. (2024); Liu et al. (2023a); Bai et al. (2023) demonstrate that increasing data granularity is crucial for enhancing the instructional capabilities of models while mitigating hallucinations. For example, InstructBLIP Dai et al. (2024) consolidates 13 diverse multimodal datasets spanning tasks like image captioning, reading comprehension from captions, and image-based question answering to strengthen the model's instructional performance. LLaVA-v1.5 Liu et al. (2023a) refines this approach by selecting four academic datasets from those used by InstructBLIP and transforming them into conversational formats using GPT OpenAI (2023), as outlined in Liu et al. (2023b).

**Adapter module.**   The adapter is critical in aligning the two modality encoders Li et al. (2023a), this adapter learns to prompt the language decoder from the visual embedding and thus an essential step for generating accurate and visually grounded responses. Recent work Li et al. (2023a); Dai et al. (2024); Liu et al. (2023a) has modified adapter architectures to enhance the model's instructional capabilities. InstructBLIP, an extension of BLIP-2, incorporates the question into its Q-Former (Adapter) module to select the most relevant visual features, enabling more accurate answers.

**Improving Visual Grounding.**   IT-VLMs typically use CLIP-based models as visual encoders Sun et al. (2023); Radford et al. (2021). However, recent studies Tong et al. (2024); Kar et al. (2024); Villa et al. (2024) reveal that CLIP-based encoders struggle to capture fine-grained visual details, resulting in visual representations that often fall short for generating grounded and accurate text responses. To address this, works such as Tong et al. (2024); Kar et al. (2024) propose ensembles of visual encoders that leverage the strengths of multiple models to enrich visual representations. While effective, these approaches do not scale well, substantially increasing the computational demand. In contrast, we introduce EAGLE, a training strategy that enhances fine-grained representation in CLIP-based encoders without sacrificing their zero-shot capabilities. Our approach allows for seamless replacement of default visual encoders in IT-VLMs with their EAGLE-tuned counterparts, requiring no architectural modifications, additional encoders, or alignment training. EAGLE demonstrates a significant reduction in hallucinations across 6 state-of-the-art IT-VLMs evaluated on 4 challenging benchmarks.

## 3 METHODOLOGY

The visual encoder of an Instruction-Tuned Visual Language Model (IT-VLM) consists of a Vision Transformer (ViT) architecture, which is commonly pre-trained in a Visual Language Model (VLM) using contrastive learning Radford et al. (2021); Sun et al. (2023); Zhai et al. (2023). During the VLM alignment phase, the ViT module is optimized to learn a global visual representation via a learnable token (CLS), which encodes high-level semantic features from the image. This CLS token is aligned with the embedding of the textual image description. As a consequence, VLMs do not explicitly supervise the entire feature sequence in the last layer of the ViT, thus limiting their ability to encode and localize fine-grained visual details. This design choice in VLMs has undesired effects in IT-VLMs, as the instructional models typically discard the (better aligned) CLS token and use the feature sequence from the pretrained ViT to prompt the LLM module.

### 3.1 VLM FEATURE ALIGNMENT

We quantify the difference in visual-language alignment for the CLS token and the feature sequence by following the protocol of Sun et al. (2023) on ImageNet-1K Deng et al. (2009). To assess the

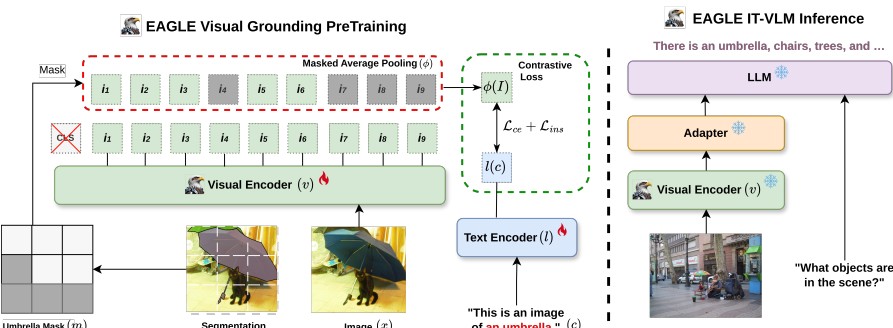

Figure 2: **Overview of the EAGLE method.** EAGLE reduces the hallucinations in IT-VLMs by improving the grounding of the image encoder. In the training phase (Left), EAGLE enhances fine-grained visual representations by employing a masked average pooling (in red dashed lines). This method selects embeddings within the feature sequence corresponding to a specific object and computes an averaged representation. Subsequently, EAGLE enforces local alignment with the language representation of the object (in green dashed lines). The resulting image encoder integrates with an IT-VLM (Right) at inference time, effectively reducing hallucinations without further tuning.

alignment of the feature sequence, we compute its average representation and use it as a surrogate CLS. As shown in Table 1a (rows 1 and 4), there is a significant gap in the alignment for the two features. For two representative VLMs Sun et al. (2023); Radford et al. (2021), the CLS token is more aligned with the language embedding, making it substantially more effective for zero-shot recognition. This finding indicates that IT-VLMs rely on a sub-optimal set of visual features.

EAGLE main goal is to tune the visual feature sequence, such that individual tokens encode a fine-grained object representation that is highly aligned with the language component. The EAGLE training, augments the capabilities of the visual encoder in two key aspects. First, the improved visual representation in the feature sequence will be aware of the object semantics and their spatial alignment in the visual data, thus reducing hallucinations in IT-VLMs. Second, the visual transformer has almost the same global feature representation and can be readily plugged into the pre-trained IT-VLM. Therefore EAGLE retains most of the zero-shot and linear probing capabilities of the original transformer (refer Table 1a and the Appendix).

### 3.2 EAGLE FEATURE ALIGNMENT

As outlined in Figure 2, EAGLE focuses on improving the feature sequence allowing the ViT module to encode fine-grained information about objects in the image using a modified contrastive learning. To generate an improved representation over the feature sequence of the ViT, we rely on image data with instance level segmentations. Formally, given a trained VLM with visual encoder $v$ and language encoder $l$, we use $v$ to obtain the feature sequence $I$ of image $x$, $I = v(x)$. $I$ has the same sequence length as the tokenized version of $x$ (*i.e* we drop the CLS token). The architecture of $l$ remains unchanged, thus the embedding of the language remains at length 1 (see blue and light green tokens in Figure 2).

For a binary segmentation mask $m$ (bottom left corner in Figure 2), we define a masked average pooling function $\phi(I, m)$ (red dashed box in 2) where the feature tokens in $I$ are set to 0 (gray tokens in 2) if they do not overlap with $m$, we perform average pooling over the non-zero tokens (green tokens). For an image with multiple segmented instances, we randomly sample one of them.

**Loss Function.** We can not directly use contrastive learning Hadsell et al. (2006) as our training objective, the standard contrastive loss considers all the elements in the batch as negatives except one. This is incompatible with our setup as the training batch might contain multiple masks with the same class $c$. We decompose our loss into two terms. $\mathcal{L}_{ins}$ which operates at image level, and $\mathcal{L}_{ce}$ which operates at batch level. Since $\mathcal{L}_{ins}$ operates at image level, that allows us to apply contrastive loss in this sub-problem, aligning the visual features of a single mask and its ground-truth language embedding while increasing the distance with the language embedding of every other class. For an image with feature sequence $I$, segmentation mask $m$, and mask's class $c$, we obtain a pair of tokens

Table 1: **Zero-Shot performance for VLMs.** Applying the EAGLE pipeline (gray rows) to a VLM leads to: (a) significantly improved alignment between visual features and language, resulting in notable zero-shot gains on ImageNet-1K (IN-1K); and (b) a substantial reduction in false positive rate (FP) when evaluated on the MS-COCO dataset.

| Model | IN-1K Zero-Shot | |
|---|---|---|
| | Cls Acc. ↑ | Sequence Acc. ↑ |
| EVA01 ViT-g-14 | **78.50%** | 57.76% |
| EVA01 ViT-g-14 (Full Fine-tuning) | 71.88% (-6.62%) | 61.21% (+3.45%) |
| EAGLE EVA01 ViT-g-14 | 76.65% (-1.85%) | **65.81% (+8.05%)** |
| OpenAI ViT-L-14-336 | **76.55%** | 0.7% |
| OpenAI ViT-L-14-336 (Full Fine-tuning) | 50.42% (-26.15%) | 39.71% (+39.01%) |
| EAGLE OpenAI ViT-L-14-336 | 71.46% (-5.09%) | **50.98% (+50.28%)** |

(a)

| Model | MS-COCO False Positives | | |
|---|---|---|---|
| | Seq. FP@1 ↓ | Seq. FP@3 ↓ | Seq. FP@5 ↓ |
| EVA01 ViT-g-14 | 24.64% | 56.68% | 68.17% |
| EAGLE EVA01 ViT-g-14 | **5.11% (-19.53%)** | **39.57% (-17.11%)** | **57.77% (-10.4%)** |
| OpenAI ViT-L-14-336 | 43.74% | 67.63% | 77.09% |
| EAGLE OpenAI ViT-L-14-336 | **29.54% (-14.2%)** | **51.35% (-16.28%)** | **63.64% (-13.45%)** |
| SigLIP-Base-16-224 | 20.80% | 55.83% | 68.05% |
| EAGLE SigLIP-Base-16-224 | **7.79% (-13.01%)** | **42.76% (-13.07%)** | **59.78% (-8.27%)** |

(b)

$\{\phi(I, m), l(c)\}$ and define:

$$\mathcal{L}_{\text{ins}} = \mathcal{L}_{\text{con}}(\phi(I, m), l(c_j)),$$

where $\mathcal{L}_{\text{con}}$ is the contrastive loss. Note that $c_j$ is composed of a prefix and the class ($j$) of a mask in natural language. We follow a prompt template where we prepend "This is an image of <mask_class>".

Our second loss function $\mathcal{L}_{\text{ce}}$ allows for multiple masks to be associated to a single class. To this end, we leverage a cross-entropy loss over a distance measure between the modalities, normalized with a sigmoid function.

$$d_j = 1 - \sigma(\phi(I, m) - l(c_j))$$

$$\mathcal{L}_{ce} = -\frac{1}{K} \sum_{j=1}^{K} c_j \log(d_j) + (1 - c_j) \log(1 - d_j)$$

where $\sigma$ is the sigmoid function, and $K$ is the total number of classes in the training set. Our final loss function integrates both losses $\mathcal{L}_{\text{ce}}$ and $\mathcal{L}_{\text{ins}}$ without extra hyperparameters.

$$\mathcal{L} = \mathcal{L}_{\text{ce}} + \mathcal{L}_{\text{ins}} \tag{1}$$

Empirically, we find that it is beneficial to have an approximate uniform distribution of training classes. Therefore, we resample such that masks with uncommon classes are more likely to be sampled than masks belonging to the most common classes.

**Training Dataset.** EAGLE requires image data with instance segmentation labels. We leverage the labeled data in the OpenImages V7 dataset Benenson & Ferrari (2022), which includes 944,037 images annotated with object segmentations distributed over 350 object classes. Although the data available in the segmented set of OpenImages is three orders of magnitude smaller than the datasets used to pre-train VLM models Sun et al. (2023); Radford et al. (2021), we show that the training strategy in EAGLE suffices to learn fine-grained visual representations at the patch level that are directly transferable to IT-VLMs. It is also important to note that OpenImages V7 lacks any instructional data, or training data for VQA tasks, therefore EAGLE does not represent any in-domain training step for IT-VLMs.

**Addressing Feature Drift in Fine-Tuning.** Empirically, we observe that our loss function enhances the visual grounding of $v$ but introduces drift from the original feature distribution of the pre-trained VLM, causing significant performance drops in zero-shot accuracy (see Table 1a rows 2 and 5). A significant feature drift will prevent the direct use of our visual encoders in VLM and IT-VLMs, and will required the joint fine-tuning (visual component and language component) to recover the multi-modal alignment.

To address this feature drift, we incorporate the Gradient Low-Rank Projection (GaLore) Zhao et al. (2024) into our training pipeline (see Table 1a rows 3 and 6). GaLore is a memory-efficient fine-tuning technique that facilitates full-parameter learning by computing a low-rank approximation of the gradient $G$ for each weight matrix $W$. This method is more memory-efficient than directly optimizing rank-decomposed weight matrices, as in LoRA Hu et al. (2022). Although GaLore supports the optimization of all parameters, we follow previous work Hu et al. (2022); Zhao et al. (2024); Yang et al. (2024) by focusing on the MLPs and linear layers within the attention-based architecture. These memory-efficient fine-tuning techniques have demonstrated effectiveness in mitigating distribution shift and catastrophic forgetting in continual learning scenarios Gao et al. (2023); Villa et al. (2023), enabling better retention of the model's original pre-trained knowledge while improving adaptation to new tasks.

Table 2: **Results on MMVP-VLM Benchmark.** We evaluate the performance (Accuracy) of EAGLE-tuned VLMs across the nine tasks defined in the MMVP-VLM Benchmark. EAGLE demonstrates consistent performance improvements on most tasks, particularly when using the feature sequence representations (SEQ). Even though the CLS token (CLS) is not directly trained in EAGLE, the tuning process still yields a performance boost for the CLS token in the EVA-01-CLIP-g-14 model. The tasks in MMVP-VLM are: ⊘ Orientation and Direction. ⚲ Presence of Specific. ⟳ State and Condition. ⇅ Quantity and Count. ⚲ Positional and Relational Context. ◉ Color and Appearance. ✿ Structural and Physical Characteristics. **A** Text. ▣ Viewpoint and Perspective.

| VLM | Feature | EAGLE | ⊘ | ⚲ | ⟳ | ⇅ | ⚲ | ◉ | ✿ | A | ▣ | Average ↑ |
|---|---|---|---|---|---|---|---|---|---|---|---|---|
| EVA-01-CLIP-g-14 | CLS | ✗ | 6.67% | 26.67% | 53.33% | 6.67% | 13.33% | 53.33% | 26.67% | 6.67% | 33.33% | 25.18% |
| **EAGLE EVA-01-CLIP-g-14** | CLS | ✓ | 13.33% | 20.00% | 53.33% | 6.67% | 13.33% | 66.67% | 26.67% | 13.33% | 26.67% | **26.67%** |
| EVA-01-CLIP-g-14 | SEQ | ✗ | 0.0% | 26.67% | 46.67% | 13.33% | 0.00% | 33.33% | 13.33% | 0.00% | 20.00% | 17.04% |
| **EAGLE EVA-01-CLIP-g-14** | SEQ | ✓ | 6.67% | 20.00% | 46.67% | 6.67% | 6.67% | 46.67% | 20.00% | 13.33% | 13.33% | **20.00%** |
| OpenAI CLIP-L-14-336 | CLS | ✗ | 0.00% | 20.00% | 40.00% | 20.00% | 6.67% | 20.00% | 33.33% | 6.67% | 26.67% | 19.26% |
| **EAGLE OpenAI CLIP-L-14-336** | CLS | ✓ | 0.00% | 20.00% | 33.33% | 6.67% | 20.00% | 20.00% | 26.67% | 13.33% | 20.00% | 15.56% |
| OpenAI CLIP-L-14-336 | SEQ | ✗ | 13.33% | 6.67% | 0.00% | 13.33% | 0.00% | 26.67% | 13.33% | 6.67% | 0.00% | 8.89% |
| **EAGLE OpenAI CLIP-L-14-336** | SEQ | ✓ | 20.00% | 33.33% | 33.33% | 0.00% | 6.67% | 20.00% | 20.00% | 6.67% | 20.00% | **22.22%** |
| SigLIP-Base-16-224 | CLS | ✗ | 6.67% | 13.33% | 53.33% | 33.33% | 13.33% | 80.00% | 33.33% | 40.00% | 26.66% | 33.33% |
| **EAGLE SigLIP-Base-16-224** | CLS | ✓ | 6.67% | 20.00% | 53.33% | 20.00% | 13.33% | 73.33% | 46.67% | 20.0% | 53.33% | **34.07%** |
| SigLIP-Base-16-224 | SEQ | ✗ | 13.33% | 6.67% | 26.67% | 6.67% | 6.67% | 66.67% | 20.0% | 13.33% | 53.33% | 23.70% |
| **EAGLE SigLIP-Base-16-224** | SEQ | ✓ | 13.33% | 6.67% | 33.33% | 6.67% | 6.67% | 80.00% | 33.33% | 20.00% | 46.67% | **27.41%** |

# 4 EXPERIMENTAL EVALUATION

We begin our empirical assessment by testing the effectiveness of EAGLE in the VLM domain. We use EAGLE to tune the pre-trained EVA-01-CLIP-g-14, SigLIP-Base-16-224, and OpenAI CLIP-L-14-336 VLMs Radford et al. (2021); Zhai et al. (2023); Sun et al. (2023) and verify if their tuned feature sequence reduces the number of hallucinated objects (false positives) when recognizing multiple objects in an image.

We validate our tuned VLMs (EAGLE VLMs) in the MS-COCO dataset Lin et al. (2014) and check for reductions in the False Positives ratio. When using the feature sequence, we calculate a global descriptor by computing the average feature representation from its tokens. For the feature sequence and CLS descriptors we rank the top K predictions (K closest language prompts), we calculate the ratio of false positives within this rank. Table 1b summarizes the results of our preliminary evaluation on VLMs. It shows that the EAGLE ViTs significantly reduce the number of false positive predictions, with up to 19.53% less false detections on the EAGLE tuned feature sequence.

**Implementation Details.** We tune the VLMs EVA01-CLIP-g-14 Sun et al. (2023), OpenAI CLIP-L-14-336 Radford et al. (2021) and SigLIP-Base-16-224 Zhai et al. (2023). The ViT in the models Radford et al. (2021); Sun et al. (2023), which are OpenAI ViT-L-14-336 and EVA01 ViT-g-14, respectively, corresponds to the visual module in MiniGPT-4 Zhu et al. (2023), BLIP-2 Li et al. (2023a), InstructBLIP Dai et al. (2024), and LLaVA-v1.5 Liu et al. (2023a). We apply GaLore Zhao et al. (2024), setting the rank to 128 and the learning rate to 4e-6. The training is performed with two A100 GPUs (80GB) with a batch size of 512. We train until convergence of the $\mathcal{L}_m$ loss. EVA01-CLIP-g-14 and OpenAI CLIP-L-14-336 converge at epoch 8, and training requires about 20 hours. In contrast, SigLIP-Base-16-224 converges at epoch 14, requiring around 27 hours of training. In Appendix A.3, we rigorously validate these hyperparameter choices, demonstrating that GaLore consistently outperforms alternative low-rank adaptation methods such as LoRA, while a rank of 128 delivers the best trade-off between efficiency and accuracy across all evaluated backbones.

## 4.1 VISUAL REPRESENTATION QUALITY

We now proceed with an in-depth empirical evaluation testing the EAGLE-tuned VLMs in the MMVP-VLM benchmark Tong et al. (2024). MMVP-VLM designs nine challenging scenarios where CLIP-based models typically fail. The failure case is designed by pairing visually distinct images that have a highly similar CLIP feature embedding. Consequently, VLMs fail to align the textual description of those image pairs. MMVP-VLM sets nine possible scenarios for visual changes in the image content: Orientation and Direction (⊘), Presence of Specific Features (⚲), State and Condition (⟳), Quantity and Count (⇅), Positional and Relational Context (⚲), Color and Appearance (◉), Structural and Physical Characteristics (✿), Text (**A**) and Viewpoint and Perspective (▣).

Table 2 presents the results of EAGLE-tuned versions of EVA-01-CLIP-g-14, OpenAI CLIP-L-14-336 and SigLIP-Base-16-224 models in MMVP-VLM. We assess performance using both the

Table 3: **Results on POPE and MMVP Benchmarks.** We evaluate the EAGLE-tuned visual encoder across multiple state-of-the-art IT-VLMs on the POPE and MMVP benchmarks. Without any additional tuning to the LLM or its fusion module, EAGLE consistently improves all metrics for every IT-VLM. Notably, LLaVA-v1.5 with a EAGLE-tuned visual encoder achieves the largest gains on MMVP, surpassing even LLaVA-v1.5 + I-MOF, which uses an ensemble of two visual encoders.

| Model | Visual Encoder | LLM | POPE | | | | MMVP |
| | | | Acc ↑ | F1 ↑ | Prec ↑ | Rec ↑ | Acc ↑ |
|---|---|---|---|---|---|---|---|
| MiniGPT-4 | EVA01 ViT-g-14 | Vicuna-7B v0 | 55.13% | 64.44% | 53.36% | 81.33% | 11.33% |
| MiniGPT-4 | **EAGLE EVA01 ViT-g-14** | Vicuna-7B v0 | **55.67%** | **64.96%** | **53.70%** | **82.2%** | **14.67%** |
| BLIP-2 | EVA01 ViT-g-14 | FlanT5xl | 69.33% | 70.90% | 67.34% | 75.06% | 14.00% |
| BLIP-2 | **EAGLE EVA01 ViT-g-14** | FlanT5xl | **71.40%** | **72.81%** | **69.40%** | **76.60%** | **15.33%** |
| InstructBLIP | EVA01 ViT-g-14 | Vicuna-7B v1.1 | 75.00% | 77.99% | 69.73% | **88.47%** | 18.00% |
| InstructBLIP | **EAGLE EVA01 ViT-g-14** | Vicuna-7B v1.1 | **77.20%** | **79.39%** | **72.44%** | 87.80% | **20.67%** |
| InstructBLIP | EVA01 ViT-g-14 | Vicuna-13B v1.1 | 64.43% | 73.73% | 59.46% | **97.00%** | **24.67%** |
| InstructBLIP | **EAGLE EVA01 ViT-g-14** | Vicuna-13B v1.1 | 67.57% | 74.92% | 61.08% | 96.80% | **24.67%** |
| InstructBLIP | EVA01 ViT-g-14 | FlanT5xl | 58.00% | 70.00% | 54.44% | **98.00%** | 18.00% |
| InstructBLIP | **EAGLE EVA01 ViT-g-14** | FlanT5xl | **59.83%** | **70.92%** | **55.58%** | 97.93% | **19.33%** |
| LLaVA-v1.5 | OpenAI ViT-L-14-336 | Vicuna-7B v1.5 | **69.33%** | **76.10%** | **62.34%** | 97.67% | 25.33% |
| LLaVA-v1.5 | **EAGLE OpenAI ViT-L-14-336** | Vicuna-7B v1.5 | 68.03% | 75.45% | 61.24% | **98.27%** | **34.00%** |
| LLaVA-v1.5 + I-MoF | OpenAI ViT-L-14-336 + DINO | Vicuna-7B v1.5 | – | – | – | – | 28.00% |

CLS token and the feature sequence, where the latter is represented by its averaged embedding and treated equivalently to the CLS token. While MMVP-VLM remains a highly challenging benchmark, EAGLE demonstrates clear improvements across most of the evaluated scenarios. In particular, the EVA-01-CLIP-g-14 model reports improvement for features extracted from the CLS token and the feature sequence. We emphasize that EAGLE's tuning strategy does not update the CLS token. Despite this, the CLS token in EVA-01-CLIP-g-14 ViT exhibits an average performance boost of 1.49%. Consistently, SigLIP-Base-16-224 gets a significant improvement for the feature sequence of 3.71% while slightly increasing the performance from the CLS token. For OpenAI CLIP-L-14-336, EAGLE tuning significantly enhances the performance of the feature sequence by 13.33%, even surpassing the CLS token's average performance. These results further verify EAGLE's effectiveness in enhancing the model's ability to capture fine-grained visual details.

## 4.2 Hallucinations in Instructional Models

We now proceed with a comprehensive empirical evaluation of state-of-the-art IT-VLMs enhanced with EAGLE on multi-modal instructional benchmarks. We select four representative hallucination benchmarks: POPE Li et al. (2023b), MMVP Tong et al. (2024), MERLIM Villa et al. (2024) and AMBER Wang et al. (2024). We run these benchmarks on six different IT-VLMs which are augmented with EAGLE-tuned visual encoders.

**Hallucination Benchmarks.** POPE Li et al. (2023b) assesses hallucination events using yes/no questions about objects in an image. For this benchmark, we focus on POPE's most challenging subset: Adversarial SEEM Zou et al. (2024) from A-OKVQA Schwenk et al. (2022), which uses the SEEM segmentation model to detect object segmentations in A-OKVQA images. In POPE, questions with "yes" answers are generated based on ground truth objects, while questions with "no" answers are formulated from the top-k most frequent objects in the dataset, which are not present in the image. POPE is a direct tool to evaluate if the visual embeddings generated by EAGLE perform better with queries addressing individual objects instead of the image's global appearance.

MMVP Tong et al. (2024) measures hallucinations using 150 image pairs and 300 corresponding (a) or (b) questions. The image pairs are designed such that they have highly similar CLIP embeddings. MMVP scores favorably only if both questions for each image pair are answered correctly. Following MMVP's methodology, we used a GPT-based scoring approach, replacing the now deprecated "GPT-4-0314" with its closest current successor, "GPT-4o" OpenAI (2023). MMVP evaluates whether or not EAGLE improves the model's recognition of fine-grained visual details.

In MERLIM Villa et al. (2024), we evaluate EAGLE using a subset of original and edited images. We target those images where an entire object category was removed from the edited image, *i.e.* there exists only 1 instance of the object in the image and was removed. This subset results in 5608 edited images and 3037 corresponding originals. MERLIM incorporates open-ended questions with equivalent meanings, to inquire about all the objects present in the image. This design choice

Table 4: **Evaluation of IT-LVLMs on MERLIM and AMBER Benchmarks.** We measure precision ($P^i_{\text{Orig}}$, $P^i_{\text{Inp}}$) of IT-LVLMs on MERLIM across original and edited image sets, where $i \in \{1, 2\}$ indexes the prompt. Prompt 1 ($i = 1$) is *'List the objects that appear in the image"* and Prompt 2 ($i = 2$) is *Enumerate the items visible in the picture"*. Our analysis targets MERLIM subsets in which an entire object category is removed from the edited images. EAGLE yields consistent precision gains across all models without extra multimodal alignment. Only LLaVA-v1.5, whose LLM is trainable, benefits from an additional alignment step with the EAGLE-enhanced visual encoder. On the AMBER benchmark, the EAGLE-encoders reduce hallucination scores (CHAIR, HAL) and increase accuracy ($Acc_D$) in the generative and discriminative tasks, respectively, underscoring EAGLE's ability to strengthen model factuality.

| Model | Visual Encoder | LLM | MERLIM | | | | | Amber | | |
|---|---|---|---|---|---|---|---|---|---|---|
| | | | $P^1_{\text{Orig}} \uparrow$ | $P^1_{\text{Inp}} \uparrow$ | $P^2_{\text{Orig}} \uparrow$ | $P^2_{\text{Inp}} \uparrow$ | Avg $\uparrow$ | CHAIR $\downarrow$ | Hal $\downarrow$ | $Acc_D \uparrow$ |
| MiniGPT-4 | EVA01 ViT-g-14 | Vicuna-7B v0 | 36.68% | 31.28% | 37.68% | 31.93% | 34.39% | 3.80 | 7.70 | **49.90** |
| MiniGPT-4 | **EAGLE EVA01 ViT-g-14** | Vicuna-7B v0 | **37.55%** | **32.73%** | **38.91%** | **33.38%** | **35.64%** | **3.50** | **7.50** | 49.50 |
| BLIP-2 | EVA01 ViT-g-14 | FlanT5xl | 57.23% | 46.84% | 58.86% | 47.47% | 52.60% | 2.80 | 5.60 | 78.50 |
| BLIP-2 | **EAGLE EVA01 ViT-g-14** | FlanT5xl | **59.95%** | **49.58%** | **63.06%** | **50.73%** | **55.83%** | **2.60** | **5.30** | **79.50** |
| InstructBLIP | EVA01 ViT-g-14 | Vicuna-7B v1.1 | 57.92% | 48.47% | 44.45% | 39.28% | 47.53% | 9.00 | 38.60 | 77.70 |
| InstructBLIP | **EAGLE EVA01 ViT-g-14** | Vicuna-7B v1.1 | **60.90%** | **50.90%** | **47.42%** | **41.86%** | **50.27%** | **7.10** | **31.90** | **78.00** |
| InstructBLIP | EVA01 ViT-g-14 | Vicuna-13B v1.1 | 36.50% | 32.02% | 32.44% | 28.43% | 32.35% | 16.70 | 65.50 | 77.80 |
| InstructBLIP | **EAGLE EVA01 ViT-g-14** | Vicuna-13B v1.1 | **40.67%** | **35.35%** | **35.28%** | **31.39%** | **35.67%** | 15.90 | 61.40 | **78.10** |
| InstructBLIP | EVA01 ViT-g-14 | FlanT5xl | 41.30% | 36.71% | 44.13% | 38.52% | 40.17% | 7.50 | 31.20 | 79.00 |
| InstructBLIP | **EAGLE EVA01 ViT-g-14** | FlanT5xl | **43.58%** | **38.78%** | **47.76%** | **42.33%** | **43.11%** | **7.00** | **28.70** | **80.50** |
| LLaVA-1.5 | OpenAI ViT-L-14-336 | Vicuna-7B v1.5 | **49.52%** | **42.57%** | 30.41% | 28.24% | 37.69% | **7.60** | **35.30** | 71.70 |
| LLaVA-1.5 | **EAGLE OpenAI ViT-L-14-336** | Vicuna-7B v1.5 | 49.17% | 42.56% | **37.91%** | **32.61%** | **40.56%** | 8.30 | 37.60 | **72.20** |

provides a more realistic scenario to assess both EAGLE's ability to capture fine-grained visual information of individual objects and the impact of the prompt phrasing.

AMBER Wang et al. (2024) comprises generative and discriminative sets designed to assess factuality and hallucination. The generative split prompts free-form image descriptions, enabling computation of hallucination metrics such as CHAIR and HAL, while the discriminative split poses yes/no questions to test fine-grained object recognition. This dual framework provides a rigorous assessment of EAGLE's ability to suppress spurious object mentions and strengthen visual grounding.

**Results on Hallucionation Benchmarks.** EAGLE improves the performance of all evaluated IT-VLMs across four benchmarks without any additional alignment or fine-tuning, except for LLaVA-1.5, whose LLM is retrained to align with the enhanced visual encoder. Results are reported in Tables 3 (POPE, MMVP) and 4 (MERLIM, AMBER). Notably, EAGLE shows a larger improvement in the more challenging benchmarks, MMVP (2.89% absolute, 15.86% relative improvement), MERLIM (2.73% absolute, 6.79% relative improvement), and AMBER (1.92% absolute, 5.51% relative hallucination reduction). MMVP requires the model to detect subtle visual variations within image pairs and then correctly align both images with the text prompts. MERLIM and AMBER check all nouns (and potential synonyms) in the responses. These results demonstrate that EAGLE encodes fine-grained visual information at the feature sequence level and reduces hallucination.

**Hidden Hallucinations.** The MERLIM benchmark introduces the concept of "hidden hallucinations" to describe seemingly correct text responses that remain unchanged after removing their visual grounding. The authors suggest that these answers are probably the result of the LLM ignoring limited visual features and simply generating consistent text Villa et al. (2024). We assess the impact of EAGLE on reducing hidden hallucinations. As shown in Table 4, EAGLE achieves significant and consistent performance gains on both original (Orig.) and edited images (Inp.). These results evidence the effectiveness of EAGLE at capturing fine-grained visual details, but also at providing more effective visual features that allow the IT-VLMs to rely less on previously generated language tokens to provide more reliable answers.

**Prompt Bias.** We further examine hallucination by analyzing instruction bias in IT-VLMs. The MERLIM benchmark provides multiple instructions for the same task, enabling an assessment of how syntactic variations on the prompt affect model responses. EAGLE–pretrained encoders do not alter the intrinsic instruction bias. As shown in Table 4, EAGLE increases precision for both prompts across all models that do not require LLM tuning, such as MiniGPT-4, BLIP-2, and the InstructBLIP family, yet it never changes their relative ranking. That is, if Prompt 1 outperforms Prompt 2 with the original ViT, this ordering persists after replacing it with a EAGLE-tuned encoder. The sole exception is LLaVA-1.5, which exhibits larger gains for Prompt 2. We attribute this to its additional LLM tuning during instructional training, allowing the model to exploit EAGLE-enhanced visual

Table 5: **Training Strategy Analysis.** We investigate the effects of EAGLE's training strategy and supervision. Notably, GaLore enables EAGLE to preserve zero-shot performance while enhancing fine-grained visual representation within the sequence embeddings, as evidenced by a substantial reduction in the false positive rate on MS-COCO.

| Visual Model | Training | Supervision | IN-1K ZS | | MS-COCO False Positives | |
|---|---|---|---|---|---|---|
| | | | Cls Acc ↑ | Seq. Acc ↑ | Seq. FP@1 ↓ | Seq. FP@3 ↓ |
| EVA01 ViT-g-14 (Baseline) | None | None | **78.50%** | 57.76% | 24.64% | 56.68% |
| EVA01 ViT-g-14 | Full-Finetune | CLS | 70.90% | 44.55% | 17.87% | 51.57% |
| EVA01 ViT-g-14 | Full-Finetune | Seq. | 71.88% | 61.21% | **3.75%** | **38.02%** |
| EVA01 ViT-g-14 | GaLore | CLS & Seq. | 74.24% | 61.66% | 4.62% | 38.14% |
| **EAGLE EVA01 ViT-g-14** | GaLore | Seq. | 76.65% | **65.81%** | 5.11% | 39.57% |

features more effectively for certain prompt structures. This reliance on LLM tuning also explains why the EAGLE-enhanced LLaVA-1.5 shows no measurable improvement on the AMBER generative set, yet achieves clear gains on the AMBER discriminative set, reflecting the model's sensitivity to the differing prompt styles.

**LLaVA Training.** The LLaVA-v1.5 model is a special case among IT-VLMs. During the instructional tuning stage, both the adapter module and the language model (LLM) are tuned. This setup introduces additional challenges for the transfer of our EAGLE model, as the tuned adapter and LLM might not directly interface with the modified feature representation. To address this, we reproduce the instructional tuning of LLaVA-v1.5, replacing the original OpenAI CLIP-L/14-336 visual backbone with our EAGLE visual encoder, following the procedure of Liu et al. (2023a). All experiments are conducted using the official LLaVA-v1.5 codebase on two A100 GPUs (80 GB each).

**Impact of Instructional Tuning.** As shown in Tables 3 and 4, our EAGLE-enhanced LLaVA-v1.5 sets a new state of the art on the MMVP benchmark, outperforming the original LLaVA-v1.5 by 8.67% and surpassing LLaVA-v1.5 + I-MoF Tong et al. (2024) by 6.0%. We follow the same instructional-tuning protocol as Liu et al. (2023a); Tong et al. (2024). Unlike LLaVA-v1.5 + I-MoF, which employs a substantially larger visual ensemble combining OpenAI CLIP-L/14-336 and DINOv2 Oquab et al. (2023), our approach relies solely on an EAGLE-enhanced OpenAI CLIP-L/14-336 with the same parameter count as the original CLIP backbone. This result highlights the ability of EAGLE to capture fine-grained visual details within its patch embeddings, achieving superior performance without increasing model size. Furthermore, our EAGLE-enhanced LLaVA-v1.5 achieves an average improvement of 2.87% on MERLIM, demonstrating that the benefits of EAGLE generalize effectively across diverse benchmarks under a unified instructional training setup.

### 4.3 TRAINING STRATEGY ANALYSIS

We conclude the empirical evaluation of EAGLE and examine the contribution of the key design choices in EAGLE's. As outlined in Table 5, the GaLore component is the most important one for maintaining zero-shot accuracy on ImageNet-1K while also enabling the sequence embeddings to capture detailed visual features, as evidenced by a substantial reduction in the false positive rate on MS-COCO. A second important design choice is to refrain from training the CLS token in favor of focusing exclusively on fine-tuning the feature sequence. Although training the CLS token further reduces the false positive rate of sequence embeddings, it compromises the zero-shot performance and reduces the generalization capability of the new model.

## 5 CONCLUSIONS

This paper presents EAGLE, a scalable and efficient approach to mitigating hallucinations in IT-VLMs. Unlike conventional methods that depend on additional instructional data, LLM fine-tuning, adapter module enhancements, or ensembling multiple visual encoders, EAGLE directly incorporates fine-grained visual grounding within the visual encoder using a dedicated tuning strategy. We demonstrate that EAGLE-tuned visual encoders integrate seamlessly into IT-VLMs without requiring additional alignment or instructional training. Our method consistently improves hallucination metrics across four standard benchmarks and six IT-VLMs, evaluating EAGLE on a diverse combination of two VLMs as visual encoders and five LLMs. Additionally, EAGLE achieves substantial gains on the challenging MMVP-VLM benchmark when applied to three state-of-the-art VLMs. This demonstrates the strong generalization capabilities of EAGLE across different VLMs and the compatibility of EAGLE-tuned visual encoders with different LLMs.

**Reproducibility Statement.** We place the highest priority on reproducibility. Upon acceptance, we will release the EAGLE-enhanced VLM model weights, along with the full training pipeline, including all hyperparameter configurations, which are listed in the Appendix. We will also provide a comprehensive evaluation suite that enforces deterministic behavior by fixing random seeds and explicitly specifying key parameters of the underlying LLMs, for example, setting `do_sample=False`, to eliminate sources of stochasticity in both VLM and IT-VLM outputs. One of our benchmarks (MMVP) relies on commercial LLMs (GPT-4o) to evaluate IT-VLM answers, so results may show minor variation across runs if the service updates or deprecates the model. Aside from these external factors, every component of our work will be fully documented and released to enable replication of all reported results.

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

# A APPENDIX

## A.1 LINEAR PROBING OF EAGLE-TUNED MODELS

We evaluate whether EAGLE effectively preserves the transfer capability of the original VLM. We compare the zero-shot and linear probing performance of EAGLE-tuned VLMs against those of the original models. As shown in Table 1a, EAGLE significantly improves the zero-shot accuracy of the feature sequence, with only minor degradation in the zero-shot accuracy of the CLS token. To further validate EAGLE's transfer capability, we analyze the linear probing performance of the visual encoder of both the original and EAGLE-tuned models. For this evaluation, we employ the official implementation provided by Sun et al. (2023). The analysis includes both the CLS token embedding and the feature sequence. For the feature sequence, we compute the logits for each individual embedding of the sequence and aggregate them by averaging to obtain the final output.

As summarized in Table 6, EAGLE-tuned models exhibit a comparable performance to the original models, even with the CLS token, which remains untrained in our entire pipeline. This demonstrates EAGLE's ability to preserve essential global features after training to capture fine-grained object information.

Table 6: **Linear Probing performance of EAGLE-tuned VLMs.** We compare the linear probing performance of the visual encoder of both the original and EAGLE-tuned VLMs. EAGLE-tuned models exhibit comparable performance to the original model, considering both the CLS Token and the feature sequence.

| Model | IN-1K | |
|---|---|---|
| | Cls Acc ↑ | Seq. Acc ↑ |
| EVA01 ViT-g-14 | **86.50%** | 85.77% |
| EAGLE EVA01 ViT-g-14 | 86.07% (-0.43%) | **85.84%** (+0.07%) |
| OpenAI ViT-L-14-336 | **85.15%** | 83.23% |
| EAGLE OpenAI ViT-L-14-336 | 83.80% (-1.35%) | **83.51%** (+0.28%) |

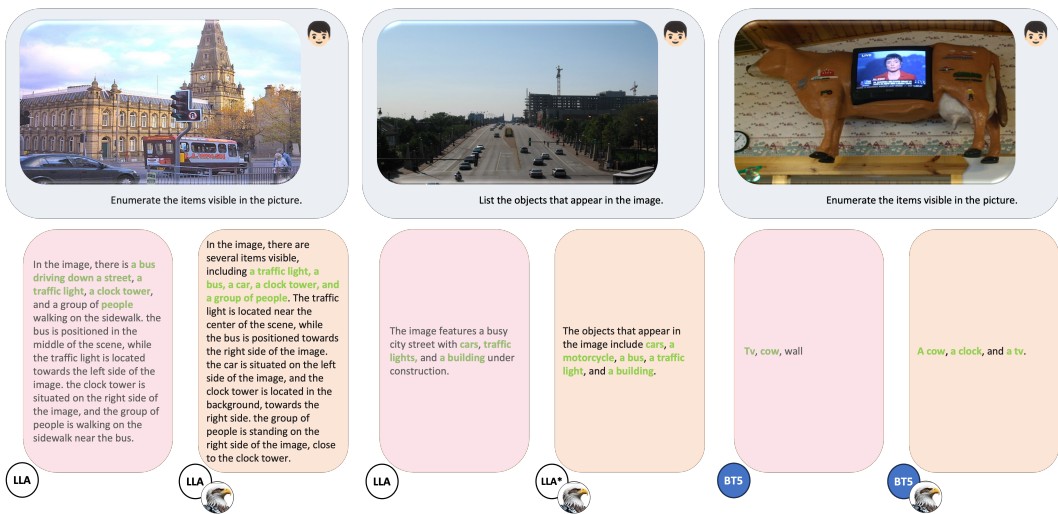

Figure 3: **Visual Examples Demonstrating EAGLE's Effectiveness in Reducing Hallucinations in IT-VLMs.** We present three additional illustrative scenarios, each featuring a question about an image processed by a specific IT-VLM using its original visual encoder (left, pink box) and the corresponding EAGLE-tuned visual encoder (right, orange box). The models evaluated include "LLA" (LLaVA-1.5) and "BT5" (BLIP-2 with FlanT5xl). EAGLE demonstrates a significant reduction in hallucinations, providing more visually grounded and reliable descriptions. Correct and incorrect predicted nouns are marked by green and red, respectively.

Figure 4: **Visual Examples of EAGLE Enhancing Visual Grounding of VLMs.** We assess the ability of two VLMs, EVA-01-CLIP-g-14 and OpenAI CLIP-L-14-336, and their corresponding EAGLE-tuned versions (blue boxes) to embed fine-grained visual details in the feature sequence, using the MMVP-VLM benchmark. Through two visual examples per model, we show that EAGLE effectively captures subtle visual information in images, enabling it to correctly align image-text pairs even when the images differ only in small, specific features. Correct and incorrect alignments are marked by green and red arrows, respectively.

## A.2 QUALITATIVE EXAMPLES

**Reducing Hallucinations in IT-VLMs.** Figure 3 shows three additional scenarios when EAGLE effectively reduces the hallucinations of the IT-VLMs. Each scenario features a question about an image by a specific IT-VLM using its original (left pink box) and EAGLE-tuned (right orange box) visual encoder. **i)** In the first example (left), LLaVA-1.5, equipped with the EAGLE-tuned visual encoder, captures more visual details than its original version, including the car and the spatial arrangement of objects in the image. **ii)** In the second example (center), LLaVA-1.5 with the EAGLE-tuned visual encoder exhibits improved visual recognition, identifying additional small elements such as the motorcycle and bus in the bottom-left and bottom-right corners of the image, respectively. **iii)** In the third example (right), EAGLE enhances BLIP-2's visual ability to detect small and partially occluded objects, enabling it to accurately identify the small clock in the bottom-left corner of the image.

**More Visual Grounded VLMs.** To further examine the impact of EAGLE on VLMs, we present visual examples from the MMVP-VLM benchmark in Figure 4 comparing the ability of the original models and their EAGLE tuned versions to correctly align image-text pairs using the feature sequence. As shown in Figure 4, EAGLE-tuned visual encoders can accurately align image-text pairs, even when the images differ in subtle, specific details. These results demonstrate EAGLE's ability to effectively enhance a VLM's capacity to embed fine-grained visual information in the feature sequence, which is critical for reducing hallucinations in IT-VLMs.

## A.3 TRAINING HYPERPARAMETERS

Table 7 summarizes the additional hyper-parameters introduced for EAGLE training, building on the original settings provided by Sun et al. (2023) for training EVA01-CLIP-g-14. The same hyper-parameters are applied to all the tuned VLMs, EVA01-CLIP-g-14, OpenAI CLIP-L-14-336, and SigLIP-Base-16-224. As detailed in Section 4, we tune these models using two A100 GPUs (80GB).

### A.3.1 EXPLORING GALORE RANKS

Table 8 shows the effect of varying the GaLore rank on the performance of EAGLE-enhanced visual encoders. Increasing the rank from 64 to 128 reduces the false positive rate of the feature sequence in MS-COCO from 30.02% to 29.54% while improving the average accuracy of MMVP-VLM from 19.26% to 22.22%. Further increasing the rank to 256 provides no additional gains, indicating a

Table 7: **EAGLE Hyperparameters.** We provide the hyperparameters we add to the original setting provided by Sun et al. (2023) for EAGLE training.

| Hyperparameter | |
| --- | --- |
| Lr | 4e-6 |
| Batch size | 512 |
| Warmup | 25000 |
| Optimizer | GaLoreAdamW |
| GaLore Rank | 128 |
| GaLore scale | 0.25 |
| GaLore Projection Type | std |
| seed | 4096 |

saturation point while increasing the number of trainable parameters. Consequently, we adopt a rank of 128 for all experiments.

Table 8: **Ablations Study on Rank and Training strategy.** We evaluate the impact of varying the GaLore rank on the fine-grained visual recognition, evaluating the false positive rate of the feature sequence on MS-COCO and its average accuracy on MMVP-VLM for EAGLE-enhanced visual encoders. Results indicate that a rank of 128 provides the best trade-off between performance and parameter efficiency, and that GaLore consistently outperforms LoRA.

| Visual Model | Training | Rank | MS-COCO False Positives | | MMVP-VLM |
| --- | --- | --- | --- | --- | --- |
| | | | Seq. FP@1 ↓ | Seq. FP@3 ↓ | Seq. Avg Acc ↑ |
| EAGLE OpenAI CLIP-L-14-336 | GaLore | 64 | 30.02% | **51.12%** | 19.26% |
| EAGLE OpenAI CLIP-L-14-336 | GaLore | 128 | **29.54%** | 51.35% | **22.22%** |
| EAGLE OpenAI CLIP-L-14-336 | GaLore | 256 | 29.38% | 51.35% | **22.22%** |
| EAGLE EVA01 ViT-g-14 | LoRA | 128 | 9.04% | 43.39% | 15.56% |
| EAGLE EVA01 ViT-g-14 | GaLore | 128 | **5.11%** | **39.57%** | **20.00%** |

## A.3.2 LoRA vs GaLore

As shown in Table 8, the GaLore training strategy substantially outperforms LoRA at the same rank of 128, reducing false positives rate of the feature sequence on MS-COCO while improving the accuracy on MMVP-VLM. Overall, these results indicate that GaLore allows better fine-grained visual recognition compared to LoRA.

## A.4 ADDITIONAL RESULTS

Table 9 reports additional AMBER results on both generative and discriminative splits. Across all IT-VLMs, the EAGLE-enhanced visual encoder improves the overall AMBER score, which aggregates performance across both sets. It also consistently lowers hallucination metrics (Hal and CHAIR) for every model except LLaVA-1.5, where gains appear only in the discriminative split. We attribute this exception to LLaVA-1.5's strong prompt-specific bias introduced during LLM fine-tuning, which can favor instructions seen in training and limit improvements on unseen generative prompts. Overall, these findings confirm that EAGLE strengthens fine-grained visual representations, enabling IT-VLMs to reduce hallucination more effectively.

Table 9: **AMBER Benchmark Results.** EAGLE-enhanced visual encoders consistently improve the overall AMBER score and reduce hallucination metrics (Hal and CHAIR) across IT-VLMs.

| Model | Visual Encoder | LLM | Amber Generative | | | | Amber Discriminative | | | | Amber Score |
| --- | --- | --- | --- | --- | --- | --- | --- | --- | --- | --- | --- |
| | | | CHAIR ↓ | Cover ↑ | Hal ↓ | Cog ↓ | Acc ↑ | Prec ↑ | Rec ↑ | F1 ↑ | |
| MiniGPT-4 | EVA01 ViT-g-14 | Vicuna-7B v0 | 3.80 | **32.40** | 7.70 | **0.40** | 49.90 | **74.40** | 37.20 | 49.60 | 72.90 |
| MiniGPT-4 | **EAGLE EVA01 ViT-g-14** | Vicuna-7B v0 | **3.50** | 32.10 | **7.50** | 0.50 | 49.50 | 73.00 | **37.80** | **49.80** | **73.15** |
| BLIP-2 | EVA01 ViT-g-14 | FlanT5xl | 2.80 | 33.00 | 5.60 | 0.30 | 78.50 | 86.20 | 80.40 | 83.20 | 90.20 |
| BLIP-2 | **EAGLE EVA01 ViT-g-14** | FlanT5xl | **2.60** | **33.50** | **5.30** | **0.30** | **79.50** | **86.80** | **81.60** | **84.10** | **90.75** |
| InstructBLIP | **EVA01 ViT-g-14** | Vicuna-7B v1.1 | 9.00 | **52.10** | 38.60 | 4.20 | 77.70 | **85.20** | 80.30 | **82.70** | 86.85 |
| InstructBLIP | **EAGLE EVA01 ViT-g-14** | Vicuna-7B v1.1 | **7.10** | 51.50 | **31.90** | **2.90** | **78.00** | 85.10 | **81.00** | **83.00** | **87.95** |
| InstructBLIP | EVA01 ViT-g-14 | Vicuna-13B v1.1 | 16.70 | **50.40** | 65.50 | 9.60 | 77.80 | 88.70 | 76.20 | 82.00 | 82.65 |
| InstructBLIP | **EAGLE EVA01 ViT-g-14** | Vicuna-13B v1.1 | **15.90** | 50.20 | **61.40** | **8.90** | **89.10** | **89.10** | **76.20** | **82.10** | **83.10** |
| InstructBLIP | EVA01 ViT-g-14 | FlanT5xl | 7.50 | 49.90 | 31.20 | 2.30 | 79.00 | 89.90 | 77.00 | 82.90 | 87.70 |
| InstructBLIP | **EAGLE EVA01 ViT-g-14** | FlanT5xl | **7.00** | **50.70** | **28.70** | **2.20** | **80.50** | **90.00** | **79.50** | **84.40** | **88.70** |
| LLaVA-1.5 | OpenAI ViT-L-14-336 | Vicuna-7B v1.5 | **7.60** | **51.60** | **35.30** | 4.20 | 71.70 | **93.30** | 61.70 | 74.30 | 83.35 |
| LLaVA-1.5 | **EAGLE OpenAI ViT-L-14-336** | Vicuna-7B v1.5 | 8.30 | 51.10 | 37.60 | **3.70** | **72.20** | 92.10 | **63.50** | **75.20** | **83.45** |

## A.5 LIMITATIONS.

EAGLE utilizes object segmentations to encourage fine-grained object representation within the feature sequence. However, the relative scarcity of image data with instance segmentations poses a challenge. To address this, we employ a parameter-efficient fine-tuning strategy, such as GaLore, which helps prevent model overfitting and preserves generalization capabilities. We encourage future work to explore surrogate sources of fine-grained supervision to further improve model grounding and performance.

## A.6 THE USE OF LARGE LANGUAGE MODELS (LLMS).

We used commercial large language models (e.g., ChatGPT) solely as editorial tools to improve the manuscript's prose and readability. Their role was limited to language editing, such as correcting grammar, improving clarity, and smoothing the flow of text, and they did not influence the research design, data analysis, or the research conclusions.

