# OpenReview forum: "EAGLE: Enhanced Visual Grounding Minimizes Hallucinations in Instructional Multimodal Models"
_ICLR.cc/2026/Conference — Submitted to ICLR 2026_

### Official Review · Reviewer_du14 · 2025-10-23

**Soundness:** 2
**Presentation:** 3
**Contribution:** 2
**Rating:** 4
**Confidence:** 3

**Summary:**

The paper introduces EAGLE (Enhanced Visual Grounding Minimizes Hallucinations in Instructional Multi-modal Models, a technique aimed at reducing hallucinations in multi-modal models (Vision-Language Models, VLMs) by enhancing the visual encoder's grounding. EAGLE modifies the vision transformer (ViT) in a post-pretraining stage to improve visual grounding without the need for additional fine-tuning or instruction training. The authors claim that this approach significantly reduces hallucinations across six different IT-VLMs and four benchmarks. The paper provide experiments to demonstrate that EAGLE enhances visual grounding, enabling more accurate and grounded language generation from images.

**Strengths:**

1. EAGLE introduces a unique solution by focusing on improving the visual encoder, rather than relying on complex changes to the language model or fusion module. This post-pretraining enhancement makes it simple to integrate into existing systems.
2. The method shows strong performance across multiple IT-VLMs and benchmarks, including some challenging scenarios such as MMVP and MERLIM, demonstrating its broad applicability and robustness.
3. EAGLE's ability to reduce hallucinations without requiring additional instructional training or fine-tuning of the entire model is a significant advantage in terms of both efficiency and scalability.

**Weaknesses:**

1. While the paper provides some experimental validation, it lacks in-depth theoretical explanation and a detailed analysis of why the proposed modifications lead to such improvements.
2. The hallucination problem may potentially be over-simplified. The solution focuses primarily on the visual component, which may not address the full range of hallucination causes. Further investigation into the interplay between the visual and language components might yield more comprehensive solutions.
3. EAGLE relies on pre-labeled datasets like OpenImages V7 for training, which limits its flexibility and raises questions about its generalizability to other types of training data or modalities.
4. EAGLE might introduce computational overhead due to the training of enhanced visual encoders. It is suggested that the authors fully explore the computational cost, especially when integrating into larger systems with more extensive multimodal components.

**Questions:**

Please refer to weaknesses.

---

> ### Author Response · Authors · 2025-12-03
>
> **1. Lack of in-depth theoretical explanation.** We appreciate the reviewer’s concern. However, Section 3 of the paper provides a detailed explanation of the motivation and mechanisms behind EAGLE. The core insight is that CLIP-like encoders supervise only the CLS embedding during training, which forces the model to compress local visual information into a single global token. This leads to patch embeddings that are semantically weak and insufficiently fine-grained, yet patch embeddings, not CLS, are exactly what IT-VLMs use as visual input. Thus, the current IT-VLM pipeline relies on suboptimal features. EAGLE directly addresses this structural limitation by supervising patch embeddings with localized semantic signals, improving fine-grained representation quality of the visual encoder while maintaining its generalization. Our empirical results across four distinct families of encoders (OpenAI-CLIP, EVA-CLIP, SigLIP, SigLIP2) consistently validate this mechanism.
>
> **2. Hallucination may be over-simplified.** We fully agree that hallucination is a multi-factor phenomenon involving the visual encoder, the adapter/projector, and the language model. EAGLE intentionally focuses on one of the most under-explored yet foundational contributors: the visual encoder. If the visual representation lacks fine-grained detail or misaligns with semantics, even the strongest LLM cannot compensate, as it receives degraded or ambiguous visual inputs.
> EAGLE is not proposed as a complete solution to all hallucination causes. Instead, it is a plug-and-play enhancement that strengthens visual grounding in any IT-VLM. Our experiments demonstrate that across multiple models, EAGLE-enhanced encoders consistently reduce hallucination, even when the adapter and LLM remain entirely frozen. This confirms that improving visual grounding is a critical and complementary direction.
>
> **3. Dependence on pre-labeled datasets limits generalizability.** We respectfully disagree with the reviewer’s concern. Although EAGLE uses a labeled dataset with segmentation masks, we rely on OpenImagesV7, which is significantly smaller and less curated than the massive proprietary datasets used to train CLIP, EVA-CLIP, SigLIP, or SigLIP2. Despite this, EAGLE substantially improves the quality of their patch embeddings. Moreover, EAGLE is not tied to OpenImagesV7. Its training data could be augmented using Grounded-SAM on a larger and unlabelled dataset.
>
> **4. Potential computational overhead.** EAGLE is explicitly designed to be lightweight. It only trains the vision encoder separately. Training is short and efficient because we focus solely on refining patch-level semantics. GaLore further reduces memory usage, enabling training with less computational resources.  Importantly, EAGLE adds zero inference-time overhead. After training, the enhanced encoder has the same computational cost as the original CLIP/SigLIP model. This makes EAGLE practical for real-world deployment, including large-scale IT-VLM systems where inference efficiency is critical.

---

### Official Review · Reviewer_rJv9 · 2025-10-23

**Soundness:** 2
**Presentation:** 2
**Contribution:** 1
**Rating:** 2
**Confidence:** 4

**Summary:**

This paper proposes a method to mitigate hallucinations in VLMs through improved visual encoder capabilities. Experimental results on MMVP and POPE benchmarks show performance gains with the enhanced visual encoder.

**Strengths:**

1.	Hallucination remains an important challenge for VLMs, and enhancing the visual encoder provides a promising direction for mitigation.

2.	EAGLE demonstrates performance gains on both MMVP and POPE.

3.	The EAGLE method is clearly presented, and Figure 2 offers a concise overview of the approach.

**Weaknesses:**

1.	It is incorrect to claim that EAGLE does not require instructional training. As stated in Line 446, applying EAGLE to LLaVA actually requires instructional training. Considering that LLaVA represents one of the most representative approaches for building VLMs, this method may not be directly applicable to modern VLMs.

2.	The experiments only demonstrate that EAGLE can reduce hallucination. Since the method introduces some misalignment with the original model, it is important to assess how other capabilities are affected. I recommend that the authors include evaluations on MMBench and OCRBench.

3.	The overall performance on MMVP is not strong, and it is inaccurate to claim that EAGLE achieves state-of-the-art results. Previous work, such as ROSS [A], achieved 49.3% by enhancing the visual encoder during the instructional tuning phase, which is substantially higher than the 34% reported in this manuscript.


[A] Wang, H., Zheng, A., Zhao, Y., Wang, T., Ge, Z., Zhang, X., & Zhang, Z. Reconstructive Visual Instruction Tuning. ICLR 2025

**Questions:**

See weaknesses.

---

> ### Author Response · Authors · 2025-12-03
>
> **1. It is incorrect to claim that EAGLE does not require instructional training.** We would like to emphasize that the EAGLE pipeline does not require any instructional data or additional training. Whether instructional data is needed when integrating EAGLE into an MLLM depends entirely on the MLLM’s training strategy. For example, in the InstructBLIP family, instructional retraining is unnecessary because the visual and text encoders remain frozen, and the model only trains an adapter. In contrast, LLaVA-1.5 finetunes both the text encoder and the adapter using instructional data for a specific visual encoder. Since the text encoder (LLM) contains a very large number of parameters, it tends to overfit to the visual encoder inputs, making it difficult for the model to immediately leverage the EAGLE-enhanced features. Consequently, in this particular case, repeating the instructional fine-tuning becomes necessary.
>
> **2. The experiments only demonstrate that EAGLE can reduce hallucination.** We respectfully disagree with the reviewer’s comment. The goal of our work is explicitly to reduce hallucination by enhancing the visual component (L020–L021). For this reason, our experiments are designed to directly evaluate whether EAGLE improves hallucination robustness, and the results consistently show that it does.
> Nevertheless, following the reviewer’s suggestion, we extended our evaluation to MiniGPT-4 and InstructBLIP (with Vicuna-7B and FlanT5xl LLMs) on the MME benchmark (14 tasks). The results indicate that while EAGLE yields minimal performance drops in perception-oriented tasks, it delivers substantial improvements in cognition-oriented tasks. In particular, EAGLE-enhanced backbones improve OCR by 2.7% and text translation by 20% on average, demonstrating that EAGLE provides meaningful benefits even on tasks for which it was not explicitly trained.
>
> | Model                           | MME Perception | MME Cognition |
> |---------------------------------|----------------|----------------|
> | InstructBLIP (Vicuna-7B v1.1)  | -0.04%         | +2.73%         |
> | InstructBLIP (FlanT5xl)        | -3.7%          | +3.54%         |
> | MiniGPT-4 (Vicuna-7B v0)       | -1.1%          | +30.6%         |
>
> **3. The overall performance on MMVP is not strong.** We respectfully disagree with the reviewer’s comment for two reasons. (1) The comparison suggested by the reviewer is not valid. The reviewer contrasts EAGLE applied to LLaVA-1.5 (CLIP-ViT-L/14@336 + Vicuna-7B-v1.5) with ROSS applied to a different and considerably stronger architecture (SigLIP-ViT-SO400M/14@384 + Qwen2-7B-Instruct). With such changes in both the vision encoder and the language model, it is impossible to isolate the source of the improvements. Indeed, the initial (zero-shot) performance in the ROSS pipeline is already 40.7, far above the LLaVA-1.5 baseline. Notably, ROSS’s relative improvement of 8.6% matches exactly the relative improvement obtained by EAGLE. (2) Under the same configuration as LLaVA-1.5 (CLIP-ViT-L/14@336 and Vicuna-7B-v1.5), ROSS [A] achieves 36.0%, corresponding to a relative improvement of 8.0%. In contrast, LLaVA-1.5 equipped with the EAGLE-enhanced CLIP-ViT-L/14@336 achieves 34.0%, with a higher relative improvement of 8.6%. This demonstrates that EAGLE delivers a stronger relative gain than ROSS [A], making it competitive and in relative terms, even superior at enhancing fine-grained recognition.
> Although ROSS reports a higher absolute score, this difference arises from discrepancies in baseline performance. In our experiments, we nearly reproduce the original LLaVA-1.5 MMVP result reported by Tong et al. (2024): 25.33% vs. 24.7%. In contrast, ROSS [A] reports an unusually high baseline of 28.0%, which naturally inflates the final absolute performance.

---

### Official Review · Reviewer_5MrE · 2025-10-30

**Soundness:** 2
**Presentation:** 2
**Contribution:** 2
**Rating:** 2
**Confidence:** 5

**Summary:**

Contrary to the traditional methods that rely on additional instructional data, fine-tuning the LLM component,
enhancing the adapter module, or ensembling multiple visual encoders to improve visual representation, this work
employs a tuning strategy to enhance fine-grained visual grounding directly within the visual encoder. Then, the
tuned visual encoder will be integrated seamlessly into an IT-VLM without requiring additional instructional training
or alignment.

To this end, this work utilized the extra dataset OpenImagesv7, which contains instance masks, to pop out instance
features from the visual encoder. Then, they will be aligned with the encoded text embeddings through contrastive
learning. Once the fine-tuned visual encoder is obtained, it will be integrated into an IT-VLM. The whole model will be
fine-tuned with parameter-efficient tuning.

This work improves hallucination metrics across four standard benchmarks and six diverse architectures, which is
also consistent with the significant improvements in the challenging scenarios for VLM proposed in the MMVP-VLM
benchmark.

**Strengths:**

1. This work addresses the hallucination issue in IT-VLM from the perspective of enhancing the fine-grained visual
grounding capability of the visual encoder directly for the first time, which sound interesting.
2. This work is proven with sufficient experiments to consolidate that is a straightforward and effective approach
that mitigates hallucinations in IT-VLMs without requiring any additional alignment or tuning.
3. The proposed method to fine-tune the visual encoder is simple and easy to follow.

**Weaknesses:**

1. All experimental settings are focused solely on three visual encoders: EVA01 ViT-g-14, OpenAI ViT-L-14-336, and
OpenAI ViT-L-14-336. Therefore, it remains to be seen whether this method can be applied to other stronger
visual encoders, such as SigClip-SO400M, to illustrate the generalizability of the approach.


2. Can the solution EAGLE be applied to the latest frameworks for addressing the hallucination issue in more
modern IT-VLMs, e.g. SigClip + Qwen (Since I can't find the corresponding experiment in both the main paper or supplementary materials)? Meanwhile, It would be better to quantitatively illustrate the extent to
which it improves the hallucination problem?


3. Please kindly evaluate the EAGLE solution on the general benchmarks, such as MMbench, MME, and ORCbench,
to estimate its performance improvement on the general leaderboard. That comparison with the latest
technique -- AnyRes proposed in Llava-Next[1] is also recommended.

4. Ross [R1] shares a similar motivation by introducing a reconstructive objective aimed at enhancing fine-grained
comprehension capabilities, which aligns with the focus on improving local visual representation in the current study.
Therefore, incorporating a discussion of Ross [R1] is encouraged.

** Minor Weakness**

1. **Missing training details**: The paper lacks details on the training setup for the visual encoder. It would be helpful to provide more details about the layers that are optimized during the training stage. Please describe what comprises of
Full-finetune and GaLore as well as more details about the training setup for each of them. Is this consistent over all
the visual encoder architectures shown in the paper? What consists of a batch during the training process? For a
given image, are the instances in the image considered as separate samples or are they considered as a single
sample? What is the effect of using multiple prompts (from CLIP based open vocabulary segmentation) on the
training process? If you are fine tuning the visual encoder, aren't you losing the zero-shot capabilities of the CLIP
model? Wouldn't it just overfil on the 350 classes of OpenImages V7? It would be great to provide evaluation on
images that have object classes not present in OpenImages V7.

[R1] H. Wang et al. Reconstructive Visual Instruction Tuning. arXiv 2410.09575

**Questions:**

1. The core methodology is essentially an extension of contrastive learning. While it effectively aligns visual patches with language via masked average pooling, the underlying concept is not particularly novel, representing more of an incremental advancement and a refinement of existing techniques rather than a novel one.

2. A critical question regarding its necessity is whether a simpler, training-free approach could achieve a similar effect. For instance, one could pre-process the image using an off-the-shelf segmentation model, tag the detected objects with their class labels, and then feed this enriched representation into the VLM. It is plausible that such a heuristic, training-free method would also significantly reduce object hallucinations.

---

> ### Author Response · Authors · 2025-12-03
>
> **1. All experimental settings are focused solely on three visual encoders: EVA01 ViT-g-14, OpenAI ViT-L-14-336, and OpenAI ViT-L-14-336.** We respectfully clarify that our paper evaluates three distinct families of CLIP-based encoders, OpenAI CLIP (OpenAI ViT-L-14-336), EVA-CLIP (EVA01 ViT-g-14), and SigLIP (SigLIP-Base-16-224), which already span a broad range of modern vision encoder architectures. Across all of them, EAGLE consistently improves patch-level representations, supporting our claim that the method is encoder-agnostic.
>
> Following Reviewer 13cn’s suggestion, we additionally experimented with SigLIP2-BASE-16, one of the latest and strongest public encoders. EAGLE again yields consistent gains, achieving a 5.18% relative improvement on the MMVP-VLM benchmark (patch embeddings) without degrading performance of the CLS embedding. We will include these results in the final version of our paper.
>
> **2. Applicability to modern IT-VLMs such as SigLIP + Qwen.** EAGLE is designed to be apply into the visual encoder, enhancing its ability to capture fine-grained visual information. It has demonstrated that it could be applied to different kind of visual encoders (Precisely the encoder families commonly used in modern IT-VLMs), like EVA-CLIP, OpenAI CLIP, SigLIP, and SigLIP2. This shows that EAGLE is fully compatible with modern IT-VLMs such as SigLIP+Qwen.
>
> **3. Evaluation on general benchmarks (MMBench, MME, ORCBench).** We appreciate the suggestion. Because EAGLE is explicitly designed to enhance fine-grained visual grounding and reduce hallucination, our primary experiments focus on hallucination-centric benchmarks. Nevertheless, following the reviewer’s request, we extended our evaluation to MME. EAGLE shows minimal performance drop on perception tasks, and significant gains on cognition tasks, including +2.7% OCR and +20% average improvement in text-translation tasks. These results demonstrate that EAGLE generalizes well and provides benefits even on tasks it was not trained for.
>
> **4. Comparison to AnyRes (LLaVA-Next).** We thank the reviewer for the suggestion. AnyRes and EAGLE are orthogonal and complementary. While AnyRes improves resolution handling via multi-scale cropping, EAGLE improves fine-grained representation learning in the visual encoder. Since the two approaches operate at different levels, they can be combined in principle. We will discuss this in the final version.
>
> **5. Discussion of ROSS (R1).** We appreciate the recommendation and will add a discussion of ROSS in the final manuscript. While both works aim to improve fine-grained visual comprehension, the methods differ fundamentally. ROSS performs reconstructive visual instruction tuning, requiring instruction data and unfreezing both vision and language modules. EAGLE is training-light, operates exclusively on the visual encoder, and does not require instruction-tuning data. It trains isolated the visual encoder. **Importantly, under the same architecture (CLIP-ViT-L/14@336 + Vicuna-7B-v1.5), EAGLE achieves a higher relative improvement (8.6%) than ROSS’s 8.0% on MMVP, while using substantially fewer compute resources and a simpler learning pipeline.**
>
> **6. The methodology is only an incremental extension of contrastive learning.** We respectfully disagree. While EAGLE is inspired by contrastive principles, its key contribution is not a minor modification to contrastive learning but a new training objective focused on patch-level semantics, which is fundamentally missing in CLIP-like encoders. Existing CLIP-style approaches supervise only the CLS token, causing the patch embeddings, the ones actually used by IT-VLMs, to remain weakly supervised and semantically shallow. EAGLE introduces an object level semantic supervision at the patch level to allow ViT module to encode fine-grained information about every object in the image. This shifts the objective from global alignment (CLS) to fine-grained local alignment, directly addressing a structural limitation of current vision encoders. Our experiments across four encoder families (CLIP, EVA-CLIP, SigLIP, SigLIP2) show that EAGLE is both effective and general, reinforcing the novelty and effectiveness of our approach.

---

### Official Review · Reviewer_13cn · 2025-10-30

**Soundness:** 2
**Presentation:** 2
**Contribution:** 1
**Rating:** 2
**Confidence:** 5

**Summary:**

This paper proposes a post-training strategy termed EAGLE for CLIP-style models. Specifically, EGALE employs a masked average pooling to a specific object and computes an averaged representation for contrastive learning. The features are expected to be more fine-grained than the original ones, and thus reduce hallucinations for downstream applications. Experiments under various evaluation protocols demonstrate the effectiveness of the proposed method.

**Strengths:**

1. The motivation is clear and somewhat reasonable.
2. The paper is easy to follow and the method seems to be simple but effective.

**Weaknesses:**

1. Limited literature review. Throughout the paper, there are no references later than 2025. Actually, works like [1, 2] also try to solve the fine-grained problem, which have been published in ICLR 2025, not to mention their follow-ups.
2. Old baselines. The comparison baselines are too old. Specifically,
    - For CLIP-style VLMs, new baselines, such as SigLIP2 and AIMv2, may partially alleviate the fine-grained problem. I doubt the effectiveness of EAGLE on these advanced backbones.
    - For LLaVA-style MLLMs, advanced methods usually *unfreeze* the visual backbone, e.g., [1] and LLaVA-NeXT. Therefore, the improvements of EAGLE may become marginal.
3. Lack of comparison with other related methods, e.g., [2].

**References**

[1] Reconstructive Visual Instruction Tuning. ICLR 2025.

[2] Diffusion Feedback Helps CLIP See Better. ICLR 2025.

**Questions:**

N/A

---

> ### Author Response · Authors · 2025-12-03
>
> **1. Limited literature review.** We thank the reviewer for suggesting additional related works. We would like to respectfully clarify that our paper already includes over 45 references, covering several strong works such as “Eyes Wide Shut” (CVPR 2024) and “BRAVE: Broadening the Visual Encoding of Vision-Language Models” (ECCV 2024, Oral).
> DIVA [2] leverages generative feedback from a text-to-image diffusion model to enhance the fine-grained visual representation of CLIP embeddings. Specifically, DIVA uses the CLS token representation projected into the multimodal space to condition the diffusion process. In contrast, EAGLE follows a parallel approach, instead of relying on the CLS token, it directly refines the patch embeddings, encouraging them to capture fine-grained visual details. EAGLE was trained with significantly fewer computational resources, only 2×A100 GPUs (80 GB) on approximately 944K instances, while DIVA required 8×A100 GPUs (80 GB) on a dataset of 3 M instances. Notably, when combined with LLAVA-1.5, EAGLE achieves a relative improvement of 8.67%, compared to 6.6% for DIVA [2] with LLAVA-1.5. This demonstrates that EAGLE represents a parallel line of work with higher improvements in fine-grained visual recognition. These results will be included in the camera ready.
>
> **2. Old baselines.** We acknowledge that the field is advancing rapidly, and therefore new models such as SigLIP2 could also be considered as baselines. However, evaluating all emerging visual encoders is infeasible due to computational constraints. In our work, we evaluate EAGLE across three representative CLIP-based encoders, OpenAI CLIP, EVA-CLIP, and SigLIP, demonstrating that EAGLE consistently enhances fine-grained visual representations across diverse architectures. Additionally, we conducted experiments with SigLIP2-BASE-16 on 2 A100 during 7 epochs, showing that EAGLE can further refine the patch embeddings even in the latest and strongest visual encoders, yielding a relative improvement of 5.18% on the MMVP-VLM benchmark (patch embeddings) without losing performance of the CLS embedding.
> Regarding LLaVA-style MLLMs that unfreeze the visual backbone, this behavior should not be interpreted as a limitation of EAGLE. Instead, it reflects the training dynamics of these models. Their end-to-end fine-tuning affects any pretrained visual encoder in the same way, whether it is CLIP, SigLIP, EVA-CLIP, or EAGLE-enhanced. In this setting, EAGLE provides a stronger fine-grained initialization that can lead to improved downstream instruction-following performance.

---

### Meta-Review · Area_Chair_u3ge · 2026-01-06

**Summary:**

The reviewers have many concerns regarding literature review, old baseline, how novel the method is, how general the method is, etc. During rebuttal, the authors partially address some of the concerns, but concerns remain.

**Reviewer Concerns:**

incomplete literature review: not addressed

old baseline: not addressed

All experimental settings are focused solely on three visual encoders: addressed

Applicability to modern IT-VLMs: not addressed

Evaluation on general benchmarks (MMBench, MME, ORCBench): not fully addressed

Comparison to AnyRes: not addressed

Discussion of ROSS (R1). not fully addressed

The methodology is only an incremental extension of contrastive learning: not fully addressed

It is incorrect to claim that EAGLE does not require instructional training: addressed

The experiments only demonstrate that EAGLE can reduce hallucination: not addressed

The overall performance on MMVP is not strong: not addressed

Lack of in-depth theoretical explanation: addressed

Hallucination may be over-simplified: not addressed

Dependence on pre-labeled datasets limits generalizability: not addressed

Potential computational overhead: not addressed

**Reviewer Scores:**

All reviewers would probably keep the score.

---

### Decision · Program_Chairs · 2026-01-26

Reject